# Evolutionary Dynamics of the Lineage 2 West Nile Virus That Caused the Largest European Epidemic: Italy 2011–2018

**DOI:** 10.3390/v11090814

**Published:** 2019-09-03

**Authors:** Carla Veo, Carla della Ventura, Ana Moreno, Francesca Rovida, Elena Percivalle, Sabrina Canziani, Debora Torri, Mattia Calzolari, Fausto Baldanti, Massimo Galli, Gianguglielmo Zehender

**Affiliations:** 1Department of Biomedical and Clinical Sciences “L.Sacco”, University of Milan, 20157 Milano, Italy; 2CRC-Coordinated Research Center “EpiSoMI”, University of Milan, 20157 Milano, Italy; 3Experimental Zooprophylactic Institute of Lombardy and Emilia-Romagna (IZSLER), 25124 Brescia, Italy; 4Molecular Virology Unit, Microbiology and Virology Department, Fondazione IRCCS Policlinico San Matteo, 27100 Pavia, Italy

**Keywords:** lineage 2 West Nile virus, evolutionary dynamics, phylodynamics, over-wintering reservoirs/vectors

## Abstract

Lineage 2 West Nile virus (WNV) caused a vast epidemic in Europe in 2018, with the highest incidence being recorded in Italy. To reconstruct the evolutionary dynamics and epidemiological history of the virus in Italy, 53 envelope gene and 26 complete genome sequences obtained from human and animal samples were characterised by means of next-generation sequencing. Phylogenetic analysis revealed two Italian strains originating between 2010 and 2012: clade A, which apparently became extinct in 2013–2014, and clade B, which was responsible for the 2018 epidemic. The mean genetic distances in clade B increased over time and with the distance between sampling locations. Bayesian birth-death and coalescent skyline plots of the clade B showed that the effective number of infections and the effective reproduction number (Re) increased between 2015 and 2018. Our data suggest that WNV-2 entered Italy in 2011 as a result of one or a few penetration events. Clade B differentiated mainly as a result of genetic drift and purifying selection, leading to the appearance of multiple locally circulating sub-clades for different times. Phylodynamic analysis showed a current expansion of the infection among reservoir birds and/or vectors.

## 1. Introduction

West Nile virus (WNV), which belongs to the *Flaviviridae* family and the *Flavivirus* genus of more than 70 species of vector-borne viruses, is a mosquito-borne virus that infects a wide range of vertebrates, including birds and mammals. It is a positive-sense, single-stranded RNA virus with a genome of approximately 11,000 nucleotides (nt). Viral RNA is translated into a single polyprotein that is processed by cellular and viral proteases in order to obtain a total of three structural proteins (corresponding to the viral core, membrane and envelope), and seven non-structural proteins (encoded by genes from *NS1* to *NS5*). In nature, the virus is maintained in a continuous enzootic cycle, with birds from the Passeriformes order as reservoirs and mosquitoes (primarily of the genus *Culex*) as vectors. Humans and horses are dead-end hosts and do not contribute to the further spread of the disease [1].

Phylogenetic analysis has established that the virus has nine lineages, but only two are human pathogens: lineage 1 (WNV-1), which is widespread in all continents, and lineage 2 (WNV-2), which used to be confined to sub-Saharan Africa but has been emerging in central Europe since the early 2000s [2], when it was first detected in Hungary in 2004, associated with sporadic cases in birds and mammals. Then it spread to eastern Austria and southern European countries [3,4], where it has caused large outbreaks such as the 2010 outbreak in Greece [5]. Following an independent introduction, another WNV-2 strain was detected in southern Russia in 2004 [6].

The first outbreak of WNV infection in Italy (caused by lineage 1) was reported in Tuscany in 1998, and led to cases of neurological disease among horses living in a large wetland area in the provinces of Florence and Pistoia [7]. For the following ten years, veterinary and human surveillance systems detected no significant circulation of WNV but, in August 2008, there was an outbreak of WNV infection that affected birds, horses and humans in eight provinces in Emilia Romagna, Veneto and Lombardy [8]. The first laboratory-confirmed human case of West Nile neuroinvasive disease (WNND) occurred in a rural area in north-eastern Italy, near the River Po [9], and phylogenetic analysis of the isolates indicated 98.8% nucleotide similarity with the strain isolated in Tuscany during 1998.

The year 2008 also saw the implementation of the Italian integrated WNV surveillance system and, despite the occurrence of a new epidemic in 2009 which, in addition to the area involved in the 2008 outbreak, also affected central Italy [10], the incidence of WNND remained relatively low (0.4 cases per million inhabitants) until 2011. However, it tripled between 2012 and 2015 (1.20 cases per million inhabitants), with a peak in 2013 (1.66 cases per million inhabitants). The mean age of the 173 patients with verified WNND between 2008 and 2015 was 73 years; 18 patients died, and the 10% case fatality rate confirmed the greater virulence of the infection among elderly people [1].

WNV-2 was first documented in Italy in co-circulation with WNV-1 in 2011 [11] and, since then, it has been most frequently identified and the number of cases of infection has increased.

Between June and November 2018, a large outbreak occurred in northern Italy led to a total of 577 human cases of WNV infection, including 230 (40.0%) neuroinvasive events and a total of 42 (18.3% case fatality rate) deaths, 279 (48.0%) cases of fever and 68 (12.0%) infections of blood donors (https://www.epicentro.iss.it). The transmission season began earlier than in other years and the virus was first detected in a pool of *Culex* mosquitoes in the province of Rovigo, where the first confirmed case was reported [12].

The aim of this study was to characterise the whole genome of WNV-2 circulating in Italy in the period 2015–2018 in order to investigate its genetic diversity using selective pressure analysis and make a phylogenetic reconstruction of its epidemiological history in Italy.

## 2. Materials and Methods

### 2.1. Ethics Statement

Informed consent was obtained from the patients according to Italian law (art.13 D.Lgs 196/2003), as well as approval of our local Ethics Committee on the use of residual biological specimens (IRB Protocol 20100000348).

### 2.2. Samples and Datasets

The study was conducted using 53 newly characterised envelope gene and 26 complete genome sequences obtained from mosquitoes (*n* = 23), birds (*n* = 41), horses (*n* = 3) and human samples (*n* = 12). The sampling period was 2015–2018, and the sampling areas were Lombardy and Emilia Romagna. Appendix A summarises the data regarding the samples newly characterised in the study.

Viral RNA was extracted from cell culture supernatants or biological samples (urine or plasma) using a QIAMP viral RNA mini Kit and the automatic QIACUBE robot (Qiagen, GmbH, Hilden, Germany). It then underwent cDNA synthesis using the SuperScriptTM First-Strand Synthesis System (Thermo Fisher Scientific, Vantaa, Finland), and was amplified by means of polymerase chain reaction (PCR) using random primers as described by [13] or, when the viral load was low, 22 primer pairs amplifying 22 overlapping genome sequences. In the case of some samples, it was only possible characterise the envelope gene using two of the 22 primers pairs.

The amplicons were cleaned using the QIAquick PCR Purification kit (Qiagen, Gmbh, Hilden, Germany) in accordance with the manufacturer’s protocol, and amplified PCR libraries were prepared using the Nextera-XT Sample Preparation kit (Illumina Inc., San Diego, CA, USA) and Hamilton robotics (Microlab STAR Line).

The nucleotides were sequenced on an Illumina MiSeq sequencer (Illumina) using the Illumina MiSeq Reagent Kit v2 (300 cycles) in order to generate 151 paired-end reads. FASTQ files were generated using MiSeq Reporter (Illumina), and the paired reads were imported to Geneious software v. R11 for analysis using an appropriate reference virus (https://www.geneious.com/).

The sequences were first trimmed to remove the primers from the read mappings, and the shortest ones were discarded. Two datasets and one subset were built. The first global dataset of European complete genome sequences contained 127 sequences and was generated by selecting sequences from Austria (AT, *n* = 20), the Czech Republic (CZ, *n* = 4), Greece (GR, *n* = 14), Hungary (HU, *n* = 3), Slovakia (SK, *n* = 4), Serbia (SR, *n* = 2), Germany (DE, *n* =3) and Italy (IT, *n* 77, of which 26 were newly characterised and are listed in Appendix A). The sampling dates ranged from 2004 to 2018, and the reference strains were retrieved from public databases (GenBank at http://www.ncbi.nlm.nih.gov./genbank/). A subset of only the Italian sequences was also generated and used to trace the phylogeographical reconstruction of the epidemic that occurred in Italy in 2018, which has never been described before.

The second dataset consisted of 130 Italian sequences of the envelope gene, including all of the 79 newly characterized sequences listed in Appendix A.

Multiple alignment of the nucleotide sequences was performed using the ClustalW algorithm implemented in Bioedit v. 7.2 software (available at http://www.mbio.ncsu.edu/BioEdit/bioedit.html).

### 2.3. Study of Recombinations and Phylogenetic Analysis

The Italian subset was phylogenetically analysed after excluding recombinant strains using the RDP4 package, which identifies potential recombinant sequences and their parents (major and minor) using RDP [14], BOOTSCAN [15], CHIMAERA [16], SISCAN [17], GENCONV [18], 3SEQ [19] and MAXCHI [20], each of which has a highest acceptable p value of 0.05 with Bonferroni’s correction for multiple comparisons The sequences indicated as being recombinant by at least three methods were excluded.

The Bayesian molecular clock method implemented in the BEAST v1.8.4 software package [21] was used to estimate the evolutionary rates of WNV. The substitution model was selected in J Modeltest [22] using the Akaike (AIC) and Bayesian information criteria (BIC) and a “decision-theoretical performance-based” approach (DT). The model selected for the first dataset was TN93 (Tamura-Nei, 93) with a proportion of invariant sites, and that selected for the envelope gene dataset was HKY (Hasegawa–Kishono–Yano).

The generalised stepping stone marginal likelihood estimator [23] was used to determine the best fitting clock and demographic model combinations. Four simple parametric models (constant, exponential, expansion and logistic population growth) and the Bayesian skyline plot (BSP), skyride and skygrid were compared as coalescent models under both a strict and a relaxed (uncorrelated log-normal, UCLN) clock [24].

The phylogenetic trees were constructed using a Bayesian Marcov Chain Monte Carlo (MCMC) approach. The chains were run for 100 million generations until reaching convergence and sampled every 10,000 steps.

### 2.4. Models for Estimating Distances

Distance estimates were obtained on the Italian not recombinant whole sequences and were calculated using MEGA 7 software, which computes the standard errors (SE) of the estimates using analytical formulas and the bootstrap method (https://www.megasoftware.net/). We calculated the mean p-distance (or nt difference) and the synonymous (syn) and non-synonymous (nsyn) distance (method of Nei Gojobori) of all of the Italian isolates, after grouping them according to the sampling year or locality. We also calculated mean genetic distances relative to the hosts (humans, birds and mosquitoes). Distances were indicated as nucleotide substitutions per 1000 sites (s/1000s).

### 2.5. Selective Pressure

Positive and negative selection were tested on Datamonkey server [https://www.datamonkey.org/] using single-likelihood ancestor counting (SLAC), fixed-effects likelihood (FEL), internal branch fixed-effects likelihood (IFEL), the mixed effects model of evolution (MEME), and fast unconstrained Bayesian approximation (FUBAR) [25].

### 2.6. Phylogeographical Analysis

The phylogeographical analysis was performed on both datasets, using the continuous-time Markov chain (CTMC) process over discrete sampling locations [26] and a Bayesian stochastic search variable selection (BSSVS) approach in order to find a minimal (parsimonious) set of rates explaining phylogenetic spread. The analyses were made using Beast 1.8.4 software (http://beast.bio.ed.ac.uk) [21,27], and showed that the relaxed clock and the least stringent demographic BSP fitted the data significantly better than the other models. The MCMC chains were run until convergence (350 million generations with sampling every 35,000 for the first dataset, and 150 million generations with sampling every 1000 for the second), which was assessed on the basis of effective sampling size (ESS ≥200) using Tracer software version 1.6 (http://tree.bio.ed.ac.uk/software/tracer/).

The obtained trees were summarised in a maximum clade credibility (MCC) tree using the Tree Annotator program after a 10% burn-in. The estimates of the time of the most recent common ancestor (tMRCA) were calculated as the number of years before the most recent sampling dates (2018). The branches of the tree were labelled using different state colours on the basis of the most probable location.

### 2.7. Bayesian Coalescent and Birth-Death Skyline Analyses

To reconstruct the evolutionary dynamics of WNV-2, we used two Bayesian skyline plots: the coalescent model, which makes it possible to estimate the changes in the effective population size (Ne) over time, and the birth-death skyline model that directly infers changes in the effective reproductive number (Re) and other important epidemiological parameters such as the death/recovery rate (becoming non-infectious) [28]. Both analyses were made using Beast v. 2.48 software and a GTR (General Time-Reversible) +gamma substitution model with a previously independently estimated fixed substitution of 4.5 × 10^−4^ subs/site/year [29].

In the coalescent analysis, the number of intervals for the effective population and group sizes was set at four.

The birth-death analysis made use of a contemporary BD skyline model with a total of one, four and five intervals, and a log-normal prior for the reproductive number (R) with a mean value (M) of 0.0 and a variance (S) of 1.25. The becoming uninfectious rate was a normal prior with M = 27 and S = 5.0 (95% confidence interval 18.8–35.2, corresponding to an infectious period of between 10.4 and 19.4 days). The sampling probability was estimated using a prior beta (1.0, 9999) corresponding to a minority of cases sampled. The origin of the epidemic was estimated by means of a normal prior with M = 6 and S = 2.0.

The MCMC analyses were run for 30 million generations, sampling every 3000 steps. Convergence was assessed on the basis of ESS values (ESS >200). Uncertainty in the estimates was indicated by 95% highest posterior density (95% HPD) intervals. The final trees were visualised using FigTree, version 1.4.

## 3. Results

### 3.1. Phylogenetic Analysis of Italian Isolates 

Twenty-six whole genomes of WNV-2 were obtained from human, avian and equine samples and mosquito pools collected in northern Italy between 2016 and 2018, and were aligned with 50 entire Italian viral genomes, some of which are publicly available. To evaluate potential recombinations, the dataset was analysed using the RDP4 method, which showed the presence of one potential recombinant strain (isolate 208iBO@16) with starting and ending breakpoints at positions 2736 and 9962, corresponding to the NS1-NS5 genes. For this reason, this sequence was removed from the final dataset.

Phylogenetic analysis showed that the Italian sequences formed two well-supported clades (pp = 0.99): clade A consisted of isolates sampled in north-east Italy between 2013 and 2014, which apparently became extinct in the same years, whereas clade B consisted of isolates from north-east and north-west Italy. The main difference between the two clades was the presence of six non-synonymous substitutions (T143A, H826Y, A937V, Y2731H, N2868S, V3414A).

The sequences of clade B formed six highly significant sub-clades: two (B1 and B2) of 15 isolates, each sampled at different times and localities, mainly in north-east Italy (Veneto and Emilia Romagna); one (B5) of eight isolates sampled between 2014 and 2018 in north-west Italy (Lombardy); and three (B3, B4 and B6) of 3–5 isolates that were less persistent, and sampled mainly in north-west Italy. In comparison with the common ancestor (Ancona, 2011), sub-clades B2 and B3 showed a total of three non-synonymous substitutions (I158T in B3; A341T and I1955V in B2); the differences of all of the other sub-clades were due to the presence of synonymous substitutions. The samples obtained in 2018 belonged to sub-clades B1, B2 and B5 (Figure 1 and Table 1).

Considering only the envelope sequences, which were more numerous than the whole genomes, we confirmed that the majority of the 2018 strains belonged to sub-clades B1, B2 and B5, the last of which also included two significant groups not detected in the full genomes (Figure 2).

### 3.2. Genetic Distances 

The mean p-distance between all of the Italian isolates was 1.73 (SE 0.18) s/1000s, for an overall mean of 15.47 (SE 1.62) mutated nucleotides per genome. Synonymous substitutions were 10 times more frequent than non-synonymous mutations (overall syn mean distance: 5.44 (SE 0.71) s/1000s vs 0.51 (SE 0.1) s/1000s nsyn distance).

There was a significant difference in the heterogeneity of the two clades: the isolates in clade A were more homogenous, with an intra-group mean distance of 1.08 (SE 0.16) vs 1.84 (0.22) s/1000s in clade B, corresponding to a mean difference of 9.67 (SE 1.31) vs 16.49 (SE 1.62) mutant nucleotides. The divergence between clades A and B was 1.71 (SE 0.18) s/1000s, which was largely due to synonymous substitutions (mean syn distance 5.36 s/1000s vs 0.52 s/1000s nsyn distance) (Table 2).

In clade B, the mean genetic distance increased in proportion to the distance between localities: in particular, the mean p-distance of the isolates obtained in localities with a shorter distance than the median (111 km) was 1.6 (SE 0.2) s/1000s, whereas that of the isolates obtained in localities with a longer distance than the median was 2.0 (SE 0.4) s/1000s (t test *p* < 0.001).

Figure 3 shows the correlations between the spatial and genetic distances of each isolate (Pearson’s r = 0.645; *p* < 0.05).

Moreover, stratifying the data according to the sampling year, the mean p-distance within each group grew longer over time, with the greatest divergence being observed in the years 2017 and 2018. The mean genetic distance between years also increased over time, but to a less significant extent (Table 3 and Figure 4).

Mean genetic distances in relation to the host were calculated after dividing the hosts into three groups (humans, birds and mosquitoes). The single isolate from horse was excluded from the analysis. The proportion of substitutions was higher among humans (mean p-distance 1.9 s/1000s, SE ±0.2) than among birds (1.5 s/1000s, SE ±0.2) or mosquitoes (1.2 s/1000s, SE ±0.2; t test p = 0.019). The mean distances between the different host species were similar to the within-species differences, with no significant differences (Appendix A).

### 3.3. Differences in Amino Acids 

Considering only the non-synonymous mutations and comparing the Italian genomes with the common ancestor (Ancona, 2011), there were 29 amino acid substitutions affecting different viral genes, 20 of which were observed in three or more isolates. Seven mutations affected NS5, five NS1, two NS3, three prM, and one ENV, NS2A and NS4A. Excluding the seven substitutions in all but one of the isolates (the ancestor), the most frequent of the remaining 13 were N2868S and V3414A (both in protein NS5), which were present in 24% of the isolates, followed by the Y2731H (NS5), A937V (NS1), H826Y (NS1) and T143A (PrM), which were present in 23% of the isolates. All of them differentiated clades A and B.

The analysis of site selection showed that three substitutions (positions H2571R, Y2731H, N2868S) were under significant positive selection supported by only one method (IFEL): all of the sites under positive selection were in the gene NS5 (Table 4).

Negative selection was identified by means of the FEL, IFEL and SLAC methods in 48, 8 and 13 codons, respectively.

### 3.4. Phylogeographical Analysis of the European and Italian WNV-2 Clades

Phylogeographical analysis of all of the WNV-2 genomes with 50 European isolates retrieved from public databases showed that the Italian isolates segregated significantly from all of the other European strains, and formed a large clade including all of the sequences sampled between 2013 and 2018. The only exception was Ancona 2011, which was at the outgroup of the Italian clade. The tMRCAs of the dated tree suggested that the divergence between clades A and B occurred between 2010 and 2012, and the phylogeographical analysis showed that the most probable locations of origin were Mantua (state posterior probability [stpp] for the entire Italian clade = 0.24, and for clade A = 0.47) and Pavia (stpp for clade B = 0.42); see Figure 5.

### 3.5. Phylodynamics (Coalescent and Birth-Death Analyses)

Birth-death and coalescent skyline plots made it possible to see the trend of infectious transmission events in parallel with that of the basic reproduction number (R0) of the clade B.

Figure 6 shows the Bayesian skyline plot of the effective population size (Ne) estimates and the birth-death skyline plot of the effective reproduction number (Re) estimates. The Bayesian coalescent reconstruction clearly shows the increase in Ne, which began in 2016 and reached log 10 in 2017 and log 100 in 2018. Likewise, the curve of mean Re values and 95%HPD using five intervals showed an increase starting from 2015, reaching 1.1 in 2016 and subsequently continuing to increase until 2018 (Re = 1.41, 95%HPD 1.14–1.80).

It was estimated that the origin of the epidemic was 4.16 years ago (95%HPD 2.2–6.0). The rate of becoming non-infectious was estimated to be 23.14 (95%HPD 12.7–33.7) 1/years, which corresponds to a mean infectious period of about 15.8 days (range 11–29).

## 4. Discussion

WNV infection was first detected in Italy in 1998, when it was restricted to animals in Tuscany [7], and then reappeared in 2008, when it affected both animals and humans living in the area of the River Po [30]. Subsequently, the WNV surveillance system recorded the constant and progressively increasing circulation of the virus every year up to the widespread epidemic of 2018. All of the early human and animal infections were caused by lineage 1 WNV (genotype 1a) [1], but, in 2011, lineage 2 was first isolated in a patient with fever living in Ancona, and was subsequently detected in other humans [11] and animals [31,32], and, since 2013, it has been found in almost all positive human and animal samples [1].

To reconstruct the evolutionary history of WNV-2 in Italy, we newly characterised the whole genome of 26 Italian isolates obtained from humans, mosquito pools, horses and birds in northern Italy using random primers for amplification as described by Djikeng et al. [13]. The new genome sequences were then aligned with 50 other Italian WNV-2 sequences partially retrieved from public databases, and compared with the whole genome of the ancestral 2011 isolate from Ancona [33]. This comparison confirmed the initial existence of at least two simultaneously circulating viral strains originating in 2010 and 2012 both diverged from a central region of the Po Valley, one towards north-east Italy and the other towards the north-west reaching the most western regions of Italy. Interestingly, the first strain apparently disappeared in 2013 [29].

The present study extends our observations to 2018 and the occurrence of the largest WNV epidemic ever seen in Italy and Europe, and makes a more detailed analysis of the genotypical characteristics of the virus.

In comparison with the prototype genome, clade A (corresponding to the previously described as the eastern strain) had 12 amino acid mutations (at positions 139, 143, 937, 978, 1335, 2210, 2731, 2756, 2868, 2975, 3179 and 3414) and 42 different nucleotides, whereas clade B (the previous western strain) showed only one amino acid substitution (at position 826) and 35 nucleotide substitutions. However, our findings only allow us to make hypotheses concerning the possible origin of these two strains from the virus isolated in 2011 after a previously hidden sylvatic cycle that was sufficiently long to allow the accumulation of such a large number of differences or whether the considerable divergence of clade A was due to at least two strains entering Italy over time.

The extinction of the A strain in 2013–2014 is supported by the fact that it was not detected in any of the samples available for this study, including those collected in 2018 during the largest epidemic of WNV infection ever occurring in Italy. Further genome characterisation of the strains detected in north-east Italy may be able to confirm this observation.

On the contrary, clade B originated in 2011–2012 and progressively diverged until at least 2018, as phylogenetic analysis revealed that it gave rise to several highly significant sub-clades with different spatial distributions and varying persistence over time. Sub-clades B1 and B2 (prevalent in the north-east) and B5 (prevalent in the north-west) still persist, including all of the strains isolated in 2018, whereas sub-clades B3, B4 and B6 were more limited in size and distribution, and their temporal clustering suggests more limited persistence such as that characterising clade A.

Interestingly, (and like Armstrong’s study of WNV-1 in the USA [34]), we found one recombinant strain among the clade A and clade B strains. It has been reported that the recombination of WNV is relatively rare [35] and does not contribute significantly to its genetic variation, and so the recombinant strain was removed from the alignment used for the selection and phylodynamic analyses.

In relation to clade B alone, there was a significant positive correlation between the mean within- and between-group genetic distances of the whole genomes and time and space, which indicates that the Italian lineage has become increasingly phylogenetically complex as a result of the continuous appearance of new locally circulating strains with the passing of time. This suggests that the evolution of WNV-2 in Italy has been driven more by stochastic elements, such as genetic drift, and less by selection [36]. Moreover, the fact that most of the mutations distinguishing the viral isolates were synonymous suggests the presence of purifying negative selection, with only a few sites undergoing significant positive selection. Most of the non-synonymous substitutions were observed between clade A and the ancestral genome Ancona 2011, whereas the single amino acid mutation in NS1 differentiating clade B from the prototype isolate, was not due to positive selection pressure.

All of these data suggest that WNV-2 entered Italy in 2011 as a result of one or two penetration events that gave rise to two circulating clades, one of which persisted until at least 2018.

Various hypotheses have been proposed to explain the appearance of the infection every year in temperate zone: one suggests that it is due to the annual migration of birds, whereas others suggest that the virus is capable of surviving the winter. Our phylogeographical analysis showed that the strains identified in Italy have always segregated from those observed in other European countries, even during the 2018 epidemic. Together with the continuous persistence of some sub-clades, this suggests the presence of endemic clades and supports the hypothesis of local over-wintering in Italy [37] more than that of the annual reintroduction of the same viral strain. However, whether over-wintering is due to vertical transmission of the infection among mosquitoes [38] or bird-to-bird local transmission during winter [39] remains to be clarified.

To investigate the temporal trend of the WNV-2 epidemic, we analysed the clade B sequences using the Bayesian coalescent and birth-death skyline models [28]. The Bayesian skyline plot showed a sharp 2-log increase in the size of the viral population between 2016–2017 and 2017–2018, thus suggesting that the infection expanded due to an increase in animal reservoirs or the availability of vectors during this period. In accordance with these observations, the estimated value of Re increased to >1 between 2015 and 2016, and peaked between 2017 and 2018.

These very interesting data require interpretation. It is well known that humans and horses are dead-end hosts in the ecological/epidemiological cycle of WNV and play no role in the chain of infection: it can consequently be assumed that R0 in humans is equal to 0 [40]. Under this condition, the size of the outbreak was largely due to the number of introductions from the reservoir (spillover) [40], and so the observed increase in Re (and also the effective number of infections) depends on the vector or reservoir, and therefore indicates an expansion of the infection in one or both of these compartments in the transmission chain.

Interestingly, the Italian integrated WNV surveillance system reported an increase in the number of infected resident and wild birds during the last three years, including magpies, carrion crows, blackbirds, little owls, and wood pigeons (Integrated surveillance of WNV and Usutu virus, bulletin n. 18, 15/11/2018). Whether this was due to a growth in the population of avian reservoir species, the introduction of new susceptible individuals or species, or an increase in the availability of vectors is not clear but, in any case, the estimated Re of >1 suggests that the infection is well established in reservoir species. An expansion in the infected reservoir could ultimately cause an increase in infections among dead-end hosts, and thus induce ever larger outbreaks especially when climatic conditions are particularly favourable to vector abundance [41], as possibly happened in the case of the 2018 epidemic.

Our findings once again demonstrate the usefulness of phylogenetic and bioinformatic methods that allow the more detailed monitoring of phenomena that are difficult to detect because they occur in the wild, but which may have a significant impact on human health. The birth-death model allowed us to estimate Re and its changes over time in a setting in which it is difficult or impossible to make a direct estimate [40].

## Figures and Tables

**Figure 1 viruses-11-00814-f001:**
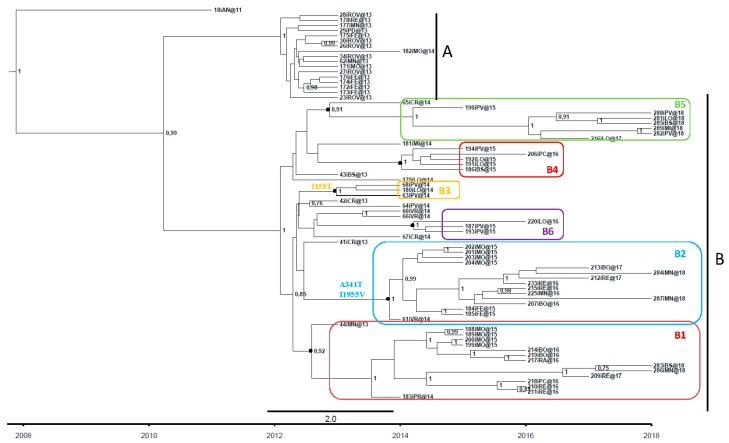
Analysis of the Italian isolates. Clades and subclades are identified respectively by black vertical lines and coloured rectangles.

**Figure 2 viruses-11-00814-f002:**
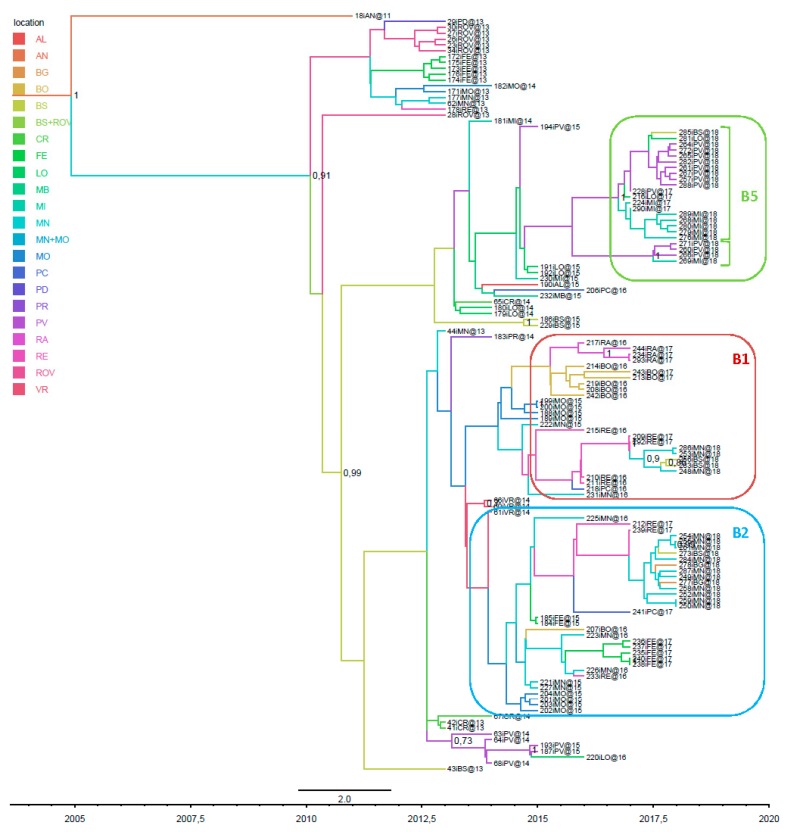
Phylogeographic analysis of 130 Italian sequences of the envelope gene. The branches of the maximum clade credibility (MCC) tree are coloured on the basis of the most probable location of the descendent nodes. The numbers on the internal nodes indicate significant posterior probabilities (pp > 0.7), and the scale at the bottom of the tree represents calendar years. The main geographical subclades, B1, B2 and B5, are highlighted.

**Figure 3 viruses-11-00814-f003:**
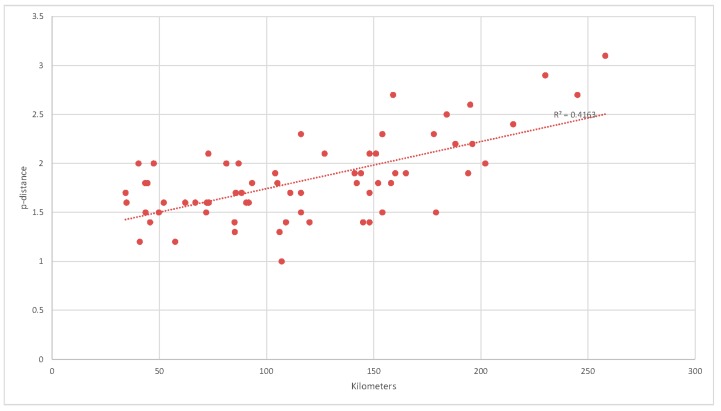
Analysis of the correlation between spatial and genetic distances of each isolate in clade B. The figure shows a growth of the genetic distance in accord to the spatial one.

**Figure 4 viruses-11-00814-f004:**
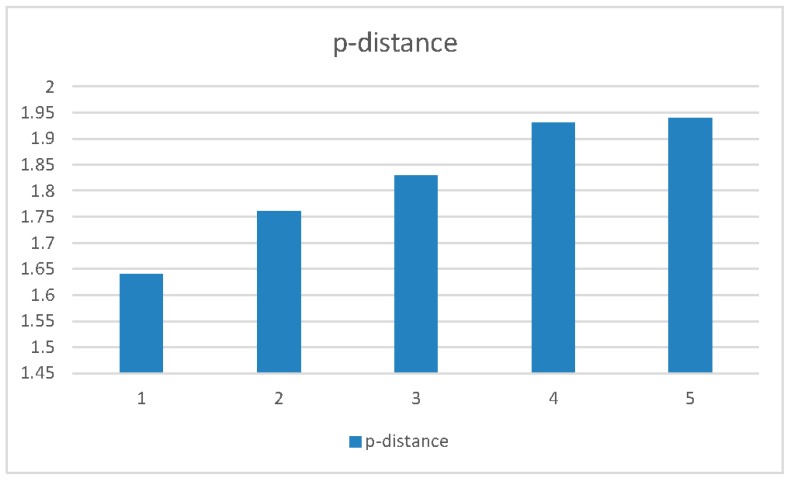
Mean p-distance between years 2013 and 2018 in clade B Italian isolates. The mean genetic distance within each group grows longer over time with the greatest divergence in the years 2017 and 2018.

**Figure 5 viruses-11-00814-f005:**
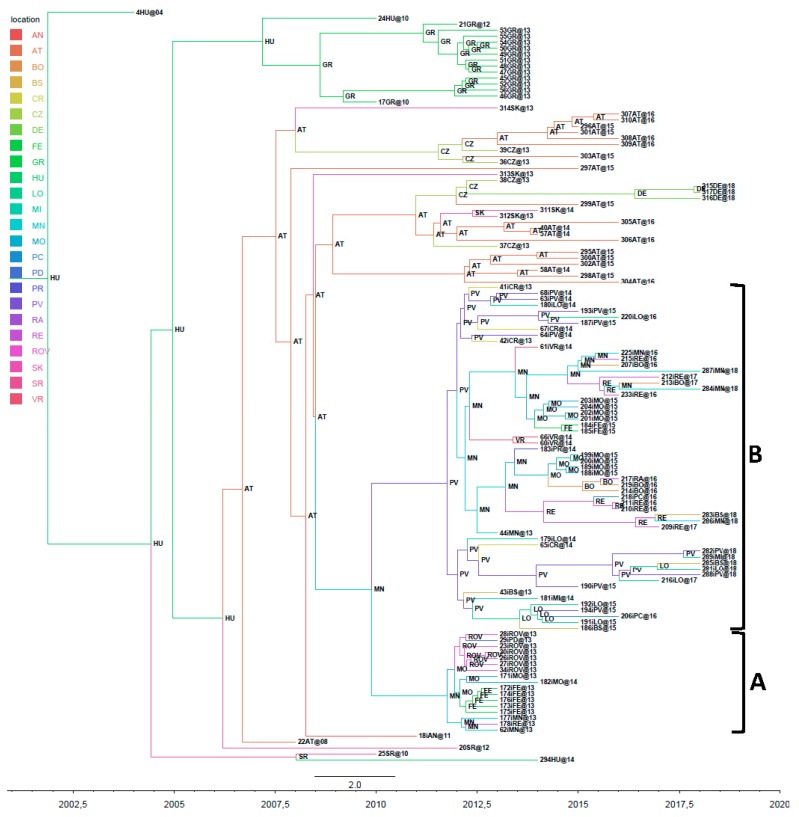
Analysis of 127 European complete genome sequences. The branches of the maximum clade credibility (MCC) tree are coloured on the basis of the most probable location of the descendent nodes. The numbers on the internal nodes indicate significant posterior probabilities (pp > 0.7), and the scale at the bottom of the tree represents calendar years. The Italian clades are highlighted.

**Figure 6 viruses-11-00814-f006:**
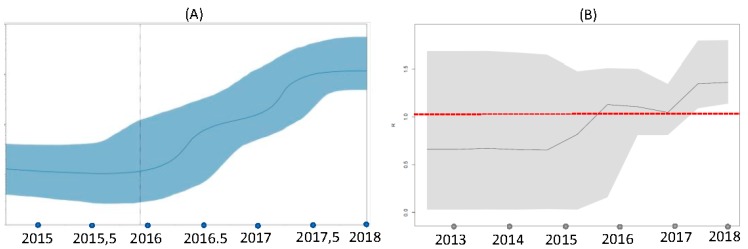
Coalescent (**A**) and birth-death skyline plot (**B**) of the WNV-2 outbreak in Italy. The Bayesian coalescent reconstruction shows the increase in Ne from 2016 to 2018. Likewise, the curve of mean R values and 95%HPD using five intervals for Re shows an increase starting from 2015 until 2018. Red dotted line indicates R = 1 level.

**Table 1 viruses-11-00814-t001:** Number of isolates, sampling years and sampling locations of the main Italian clades and subclades from 2013 to 2018.

WNV Group	2013		2014		2015		2016		2017		2018	
	N^1^	L^2^	N	L	N	L	N	L	N	L	N	L
**A** (17)	16	VE^3^ LO^4^ E^5^	1	E	-	-	-	-	-	-	-	-
**B** (49)												
**B1** (15)	1	LO	1	E	4	E	6	LO-E	1	E	2	LO
**B2** (15)	-	-	1	VE	6	E	4	LO-E	2	E	2	LO
**B3** (3)	-	-	3	LO	-	-	-	-	-	-	-	-
**B4** (5)	-	-	-	-	4	LO	1	LO	-	-	-	-
**B5** (8)	-	-	1	LO	1	LO	-	-	1	LO	5	LO
**B6** (3)	-	-	-	-	2	LO	1	LO	-	-	-	-

N^1^= Number of isolates. L^2^= Locality. VE^3^= Veneto. LO^4^ = Lombardy. E^5^= Emilia Romagna.

**Table 2 viruses-11-00814-t002:** **Mean** genetic divergence within and between A and B clades in terms of synonymous and non-synonymous substitutions.

Within	A	B	Total
Mean p distance (SE)	1.08 (0.16)	1.84 (0.22)	1.73 (0.18)
n° of differences (SE)	9.67 (1.31)	16.49 (1.62)	15.47 (1.62)
Synonymous (SE)	3.16 (0.65)	5.93 (0.76)	5.44 (0.71)
Non synonymous (SE)	0.42 (0.10)	0.48 (0.12)	0.51 (0.1)
**Between**			
Mean p-distance (SE)	-	1.71 (0.18)	-
n° of differences (SE)	-	15,32 (1.89)	-
Synonymous (SE)	-	5.36 (0.83)	-
Non synonymous (SE)	-	0.52 (0.12)	

**Table 3 viruses-11-00814-t003:** Mean p-distance within and between years 2013–2018 in clade B Italian isolates. The table shows an increase of the mean genetic distance between years over time, but to a less significant extent compared to the distance within years.

**Within**	
**Years**	**Mean P-Distance (SE)**
2013	0.55 (0.18)
2014	1.1 (0.12)
2015	1.49 (0.19)
2016	1.61 (0.22)
2017	2.38 (0.33)
2018	2.54 (0.25)
Total	1.67 (0.18)
**Between**	
**Δ Years**	**Mean P-Distance (SE)**
1	1.64 (0.56)
2	1.77 (0.5)
3	1.83 (0.48)
4	1.94 (0.27)
5	1.94 (0.27)

**Table 4 viruses-11-00814-t004:** Amino acid changes in the Italian isolates and analysis of the selective pressure.

NO OF VARIANTS (/76)	%	AA CHANGES	POSITION	GENE	METHOD
75	0.99	V -> A	416 (139)	prM	-
17	0.23	T -> A	427 (143)	prM	-
3	0.04	I -> S	473 (158)	prM	-
15	0.2	A -> T	1021 (341)	ENV	-
17	0.23	H -> Y	2476 (826)	NS1	-
5	0.07	I -> V	2509 (837)	NS1	-
4	0.05	I -> F	2740 (914)	NS1	-
17	0.23	A -> V	2810 (937)	NS1	-
75	0.99	V -> A	2933 (978)	NS1	-
75	0.99	S -> C	4003 (1335)	NS2A	-
3	0.04	R -> G	5263 (1755)	NS3	-
14	0.18	I -> V	5863 (1955)	NS3	-
75	0.99	T -> I	6629 (2210)	NS4A	-
6	0.08	H -> R	7712 (2571)	NS5	**IFEL**
17	0.23	Y -> H	8191 (2731)	NS5	**IFEL**
75	0.99	S -> G	8266 (2756)	NS5	-
18	0.24	N -> S	8603 (2868)	NS5	**IFEL**
75	0.99	H -> R	8924 (2975)	NS5	-
75	0.99	A -> T	9535 (3179)	NS5	-
18	0.24	V -> A	10241 (3414)	NS5	-

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
