# Peer review of "Evolutionary Dynamics of the Lineage 2 West Nile Virus That Caused the Largest European Epidemic: Italy 2011–2018"

_viruses, 2019, doi:10.3390/v11090814_

Round 1
Reviewer 1 Report
The manuscript is a well-written analysis of the evolutionary dynamics of lineage 2 West Nile virus in Italy. The authors analyze 26 whole genomes from WNV2 and 50 Italian viral genomes. Phylogenetic analysis shows two clades (A&B) with emergence and establishment of specific subclades later in the epidemic. Genetic distance correlates with spatial distance in clade B and time contributed less to p-distance. Interestingly, isolates obtained from different hosts did not significantly contribute to p-distance and most variation was synonymous mutations with little evidence of selection pressure. The authors conclude that the current data provides insight into the WNV outbreak in Italy and then ongoing evolution of clade B strains is likely a stochastic event rather than driven by selection. They also show that viral population growth may be associated with expanded reservoirs. Overall, the manuscript provides new insight into the development of viral populations and changes over time. As a minor comment, some of the numbers included in each analysis don’t necessarily add up to the described number of isolates used (26 whole genomes and 50 Italian viral genomes) but methods and table S1 provide different numbers. It would be helpful to know the number of isolates included in each analysis. For example, how many human, bird, mosquito isolates were analyzed in Table S2. Similarly, it is not clear what sequences are used for each analysis, Envelope sequences at times and whole genome at other times. It would be helpful for the reader to be more explicit in describing what sequences were used for each analysis.
Author Response
We would like to thank the referee for his positive consideration in our study. Point by point replies follow.
In the Table S1 are summarized only the sequences newly characterized, excluding those retrieved from public databases (50 of which were Italian whole genomes).
For a better comprehension, Table S1 was modified specifying the portion characterized (envelope gene or entire genome) and the number of isolates for each host group was added to Table S2.
Partial envelope sequences were used only for the phylogeographical analysis. All the other analyses were performed on the whole sequences (first dataset).
We described in more detail the isolates used for the different analyses in the revised manuscript.

Reviewer 2 Report
This is a well-written manuscript on the genetic diversity of WNV strains circulating in Italy.
I suggest some minor, technical corrections:
family Flaviviridae, genus Flavivirus and genus Culex should be written in italic; please correct neuroinvasive disease; Discussion - page 11, line 318 - please correct "... for amplification as described by Djikeng et al. [13]; References - please check the journal citations (journal names should be written in uppercase letters?)
Author Response
We would like to thank the referee for his positive consideration in our study.
The text was revised according to his/her suggestions. In particular, family and genus name were written in italics, page 11, line 318 was corrected and references have been revised.
